# Advancing Brain Metastases Detection in T1-Weighted Contrast-Enhanced 3D MRI Using Noisy Student-Based Training

**DOI:** 10.3390/diagnostics12082023

**Published:** 2022-08-21

**Authors:** Engin Dikici, Xuan V. Nguyen, Matthew Bigelow, John L. Ryu, Luciano M. Prevedello

**Affiliations:** 1Department of Radiology, The Ohio State University College of Medicine, Columbus, OH 43210, USA; 2ProScan Imaging, Columbus, OH 43230, USA

**Keywords:** brain metastases, noisy student, semi-supervised training

## Abstract

The detection of brain metastases (BM) in their early stages could have a positive impact on the outcome of cancer patients. The authors previously developed a framework for detecting small BM (with diameters of <15 mm) in T1-weighted contrast-enhanced 3D magnetic resonance images (T1c). This study aimed to advance the framework with a noisy-student-based self-training strategy to use a large corpus of unlabeled T1c data. Accordingly, a sensitivity-based noisy-student learning approach was formulated to provide high BM detection sensitivity with a reduced count of false positives. This paper (1) proposes student/teacher convolutional neural network architectures, (2) presents data and model noising mechanisms, and (3) introduces a novel pseudo-labeling strategy factoring in the sensitivity constraint. The evaluation was performed using 217 labeled and 1247 unlabeled exams via two-fold cross-validation. The framework utilizing only the labeled exams produced 9.23 false positives for 90% BM detection sensitivity, whereas the one using the introduced learning strategy led to ~9% reduction in false detections (i.e., 8.44). Significant reductions in false positives (>10%) were also observed in reduced labeled data scenarios (using 50% and 75% of labeled data). The results suggest that the introduced strategy could be utilized in existing medical detection applications with access to unlabeled datasets to elevate their performances.

## 1. Introduction

Brain metastases (BM) are cancerous lesions indicating an advanced and disseminated state of disease. The early detection of BM may enable a treatment utilizing targeted radiotherapy that (1) allows for a less-invasive and less-costly procedure when compared to surgery and (2) leads to fewer adverse neurologic symptoms when compared to whole-brain radiation [1]. Contrast-enhanced 3D magnetic resonance imaging is the key modality for the detection, characterization, and monitoring of BM. However, the task can become challenging when BM lesions are very small; their low contrast and structural similarities with surrounding structures (in some slice angles) may obstruct/limit their visual detection [2].

The automated detection/segmentation of BM in MRI data via machine learning (ML) was investigated in several studies [3,4,5,6,7,8]; Cho et al. [9] provided a literature review study on the topic comparing state-of-the-art (SOTA) approaches based on the Checklist for Artificial Intelligence in Medical Imaging (CLAIM) [10] and Quality Assessment of Diagnostic Accuracy Studies (QUADAS-2) criteria [11]. The authors previously introduced a framework for T1-weighted contrast-enhanced 3D MRI [4] (analyzed among other SOTA approaches in [9]) for the detection of BM with diameters less than 15 mm to assist early detection of disease.

The noisy student (NS)-based self-training approach was first introduced in [12] to enable the usage of large amounts of unlabeled (i.e., not segmented or annotated for ML training) datasets to improve the accuracy of existing ML-based solutions. While the approach is still new as of the writing of this manuscript, it has already been utilized in various domains, including medical imaging. In [13], the NS method was employed for abdominal organ segmentation in 3D computed tomography (CT) datasets. The study utilized 3D nnU-Net [14] as both the teacher and student models, which were trained with 41 labeled and 363 unlabeled datasets. The study reported ~3% improvement in the Dice similarity coefficient (DSC) due to unlabeled data. Rajan et al. [15] used the approach for multi-label classification of chest X-ray images. In addition to classical image augmentations, the method also utilized mixup [16] and confidence-tempering regularizations for the noisy student model’s training. Their results showed that a ResNet-18 [17] trained via their scheme with 12.5k labeled and 15k unlabeled samples could outperform a similar model trained using a 138k labeled set. In [18], Shak et al. proposed an NS-based lung cancer prediction framework for CT images, where a DeepSEED network [19] was trained using labeled Lung Image Database Consortium image collection (LIDC) [20] and unlabeled National Lung Screening Trial (NLST) [21] datasets. The applicability of the NS for the segmentation of intracranial hemorrhages in CT was presented in [22]. The study included 456 labeled and 25k unlabeled head CT exams for the semi-supervised training of a PatchFCN [23], producing a ROC area under the curve (AUC) of 0.939 for the CQ500 dataset [24]. Kim et al. [25] utilized the NS for the quantification of severity and localization of COVID-19 lesions on chest radiographs. Their framework consisted of a vision transformer-based [26] backbone, trained using ~1000 labeled and ~190k unlabeled images collected from public datasets (i.e., [27,28]), and provided results comparable to those of expert radiologists. Lastly, Ref. [29] proposed a cascaded learning approach for whole heart segmentation in CT angiography data. The method employed a V-net-based backbone [30] and advanced the NS with a shape-constrained training, producing a Dice coefficient of 0.917 with only 16 labeled and 64 unlabeled 3D images.

The goal of this study is to advance the BM detection framework [4] via an NS-based self-training strategy to utilize a large corpus of unlabeled T1c data in a semi-supervised fashion. The BM detection sensitivity in connection with the number of false detections has been the main criteria for various BM detection solutions [3,4,6]. Thus, the study introduces a sensitivity-based noisy-student training approach to factor in the BM detection sensitivity. To this end, the advanced framework uses (1) the CropNet-based architectures as the teacher and student models, (2) model and data noising procedures to enable knowledge expansion, and (3) a novel pseudo-labeling strategy to factor in the BM detection sensitivity during the models’ training. The proposed models are trained in an iterative fashion using a smaller group of labeled and a larger group of pseudo-labeled data; the proposed adaptation of the detection framework is illustrated in Figure 1.

This manuscript first provides a brief overview of the BM detection framework. Then, it describes (1) the model architectures, (2) noising mechanisms, and (3) a novel NS-based semi-supervised training setup that introduces a pseudo-labeling strategy. The results section summarizes the experiments performed using two-fold cross-validation (CV) on 217 labeled and 1247 unlabeled T1c exams. It presents comparative analyses for (1) the frameworks trained with and without the given NS-based strategy and (2) hyperparameters regarding the NS-based training process. Next, the results are discussed, where the framework is compared with other SOTA approaches via metrics including the BM detection sensitivity and false BM detection count. The report concludes with a summary of the novelties of the introduced study and future work considerations.

## 2. Materials and Methods

### 2.1. BM Detection Framework Overview

The BM detection framework consists of two major stages: (1) candidate tumor detection and (2) classification of the candidates. The candidate selection procedure adapts the Laplacian of Gaussian (LoG) [31] approach for maximizing the BM detection sensitivity while minimizing the number of candidates [4]. The detected candidates are then used as centers of the volumetric region of interests (ROIs) for processing by the CropNet, a convolutional neural network, to classify a given ROI as BM or not.

The CropNet model architecture was first introduced in [4]. The model processes isotropically sampled cubic ROIs, with each voxel representing 1 mm^3^. It inherits a contracting network architecture, where (1) each resolution level is processed via a set of blocks consisting of convolution, rectified linear activation unit (i.e., ReLU), and dropout layers, and (2) the resolution downsampling is performed via max-pooling layers. The input volume’s dimensions and the network’s blocks per resolution level are denoted in the model’s name; e.g., CropNet-bX-Ymm describes a network with X blocks per resolution level that processes cubic regions with the edge length of Y mm.

The LoG approach produces thousands of candidates (i.e., ~70k) for each 3D dataset; hence, the ROIs with BM are under-represented. Thus, the framework employs a paired training strategy, where each training data batch has an equal number of positive and negative samples (i.e., ROIs with 1-BM and 0-non-BM). The binary cross-entropy loss for this classification problem is minimized during CropNet’s training.

### 2.2. Teacher–Student Models and Noising Mechanisms

During the NS training, the student model capacity needs to meet or exceed that of the teacher model to enable knowledge expansion (e.g., EfficientNet-B7 and EfficientNet-L2 were used as the teacher–student pair in [12], presenting a seven-fold network capacity scaling). Accordingly, dedicated BM classification models CropNet-b2-16mm and CropNet-b4-16mm are utilized as the teacher–student pair in this study. The network architectures for these models are shown in Figure 2.

The noising mechanisms are necessary for improving the generalizability of the neural networks, especially in limited-data scenarios [32]. They are critical for NS-based self-training strategies to enforce invariances in the decision function during the training of the student model. Both the model and data noising mechanisms are deployed: the data noising is applied via random Simard-type 3D deformations, gamma corrections, rotations, and image flips, whereas the model noising is provided via dropout layers (see Figure 3).

### 2.3. Technical Contribution: Sensitivity-Based NS Algorithm

The labeled data set is initially processed via the constrained LoG algorithm to generate BM candidates for the given data. The candidates and the manual BM annotations are then used for producing a set of ROI volumes X with their corresponding labels Y(1: BM, 0: non-BM), where X is a paired set with an equal number of positive and negative samples. The teacher model $\theta^{t}$ is trained with the data noised version of the extracted ROIs by minimizing the binary cross-entropy type loss function ℓ as:(1)$$\arg min_{\theta^{t}}\left(\ell \left(\theta^{t}\left(X_{noised}\right),Y\right)\right).$$

After the training of the teacher model, the unlabeled data is processed using the constrained LoG algorithm to generate BM candidates. As there are no annotations for the unlabeled data, the pseudo-labels need to be generated. The framework produces BM detections based on a model response threshold μ, determined using the BM detection sensitivity. Thus, the relationship between the threshold and the teacher model sensitivity (ts) can be learned from the training data. Accordingly, the sensitivity in relation to the false BM detections (generated by $\theta^{t}$ on  X) was computed, and a response threshold μ was set based on this value; for the unlabeled ROIs $\hat{X}=\left\{{\hat{x}}_{1},{\hat{x}}_{2},\cdots {\hat{x}}_{N}\right\}$, pseudo-labels $\hat{Y}$ are determined by (see Figure 4),
(2)$${\hat{y}}_{i}=\{\begin{array}{cc} 1 & \theta^{t}\left({\hat{x}}_{i}\right)>\mu ,\\ 0 & else. \end{array}$$

The student model is optimized using both labeled and pseudo-labeled data as
(3)$$\arg min_{\theta^{s}}\left(\ell \left(\theta^{s}\left(X_{noised}\right),Y\right)+\lambda \ell \left(\theta^{s}\left({\hat{X}}_{noised}\right),\hat{Y}\right)\right),$$
where λ determines the weight of the unlabeled data on the final loss. In [12], their loss equally weighted the labeled and pseudo-labeled parts of the equation by normalizing these with the corresponding sample counts. Hence, their solution could be considered as a special case of our formulation with λ=1.

After the generation of the student model, it replaces the teacher model. A new student model(s) could be trained iteratively following the same procedures. The suggested algorithm has three hyperparameters; μ, λ, and student model iterations. As μ is determined based on the teacher model’s sensitivity for the training data, the usage of a value derived for the model’s peak sensitivity is suggested. The motivation behind this choice is to allow the detection of most of the BM; even this may lead to more false-positive pseudo-labels. The alternative choice of setting μ for a lower sensitivity would lead to highly accurate pseudo-labels, whereas the challenging BM detections with low $\theta^{t}\left({\hat{x}}_{i}\right)$ are excluded from the student model’s training.

The adoption of λ value of 1.0 and performance of a single student model iteration are suggested, as each iteration requires an extensive amount of computational resources, where the resulting performance boost decays significantly after the first iteration. The effects of changing these parameters are presented via experimental studies.

### 2.4. Database

The data were collected retrospectively following Institutional Review Board Approval with a waiver of informed consent (institutional IRB ID: 2016H0084). The labeled data included 217 post-gadolinium T1-weighted 3D MRI exams (contrast agent: gadoterate meglumine—0.1 mmol/kg of body weight) and their corresponding BM segmentation masks. The exams were acquired between January 2015 and February 2016 from 158 patients; 113 patients had a single, 33 patients had two, 10 patients had three, and two patients had four imaging exams. The group had (1) a mean age of 62 (σ = 10.5) and (2) a sex ratio (males:females) of 0.89. They were selected based on finalized brain MRI reports generated by a neuroradiologist, where the exams with reports detailing primarily parenchymal enhancing metastases, ideally those undergoing surveillance for radiation therapy/radiation treatment planning, were selected. Patients with metastases larger than 15 mm in diameter, primary brain neoplasms, central nervous system lymphoma, extra-axial disease, leptomeningeal disease, or equivocally enhancing foci were excluded. The BM segmentation masks were then generated by a fourth-year radiology resident, using the finalized examination report and/or annotated Picture Archiving and Communication System (PACS) images to ensure all BM were correctly delineated. The segmentation data included 932 annotated BM, where (1) the mean number of BM per patient was 4.29 (σ = 5.52), (2) the mean BM diameter was 5.45 mm (σ = 2.67 mm), and (3) the mean BM volume was 159.58 mm^3^ (σ = 275.53 mm^3^).

The unlabeled data included 1247 post-gadolinium T1-weighted 3D MRI exams, acquired between November 2016 and December 2019 from a non-overlapping group of 867 patients (i.e., no common patients between the labeled and unlabeled data). Of these, 579 patients had a single, 208 patients had two, 68 patients had three, and 12 patients had four MRI exams. The group had (1) a mean age of 56 (σ = 14.5) and (2) a sex ratio of 1.09. These patients were selected based on the diagnosis codes for malignant neoplasm(s) and/or secondary malignant brain neoplasm(s). A selected group of information of the labeled and unlabeled datasets is summarized in Table 1.

### 2.5. Validation Metric

The average number of false positives AFP (i.e., the count of falsely detected tumor locations) in connection with the detection sensitivity (i.e., the percentage of the actual BM detected) was used as the validation metric. A tumor was tagged as detected if the framework generated a detection up to 1.5 mm apart from the tumor’s center. The metric provides a relevant measurement for the algorithm’s applicability in real-life deployment scenarios, as (1) the sensitivity of a detection system is critical, and (2) the number of false positives needs to be low to ensure the system’s feasibility for optimal clinical workflow. Accordingly, it was employed in various SOTA BM detection studies, including [3,4,6,33].

## 3. Results

### 3.1. Validation Study

The approach was validated using a two-fold CV. The CV bins were set based on patients (i.e., the data from each patient can only occur in either the training or testing bin), where each bin included the labeled data from 79 patients. For each CV-fold, with the LoG candidate detector tuned as in [4], the setup generated ~72K BM candidates and captured ~95% of the actual BM centers in the training bin. The teacher model was paired-trained using the labeled data in the training bin with its corresponding BM candidates. The teacher model’s response threshold (μ) was determined by targeting the detection sensitivity (ts) of 90% for the training bin. Next, the student model was paired-trained using (1) the labeled data in the training bin, (2) the unlabeled data that was pseudo-labeled by the teacher model, and (3) their corresponding BM candidates. The λ hyperparameter of the student model loss was set to 1.0. Finally, both the student and teacher models were tested for the testing bin (having no overlapping patients with the training bin and unlabeled data), generating the AFP versus sensitivity charts (see Figure 5). The teacher model produced AFPs of 2.95, 5.74, and 9.23 for the 80%, 85%, and 90% detection sensitivities, respectively. The student model produced AFPs of 2.91, 4.82, and 8.44 at the same sensitivity levels (please note that teacher model performance closely mimics the BM detector proposed in [4] as they both employed identical model architectures and were trained using similar labeled data). Accordingly, the AFP reduction for the 90% BM detection sensitivity was ~9% (from 9.23 to 8.44) for the introduced strategy. Figure 6 presents a set of example BM detections for the framework.

The teacher (CropNet-b2-16mm) and student (CropNet-b4-16mm) networks consisted of ~14M and ~32M trainable parameters, respectively. The networks had architectures described in Section 2.2, processing cubic ROIs with 16 mm edges. The models’ dropout rates were set at 0.15. The model optimizations were performed using the Adam algorithm [34] with (1) a learning rate of 0.00005 and (2) exponential decay rates for the first- and second-moment estimates of 0.9 and 0.999, respectively. Both models were trained using 12,000 batch-training iterations, without an early stopping condition. Each batch included (1) 150 positive and 150 negative samples for the teacher models and (2) 75 positive, 75 pseudo-labeled positive, 75 negative, and 75 pseudo-labeled negative samples for the student models. The implementation was performed using the Python programming language (v3.6.10), and the neural networks were created using the TensorFlow (v1.12.0) library. The training times for the teacher and student models were ~7.25 and ~10.5 h, respectively, using an NVIDIA Tesla V100 graphics card with 32 GB RAM.

### 3.2. Experiments with System Parameters

The introduced noisy-student-based self-training strategy has three input parameters (i.e., *μ*, *λ*, and student model iterations); the reasoning behind their default values was provided previously. Three analyses were performed to illustrate the effects of these parameters, and an additional analysis was performed to present the impact of labeled data amount in our application.

#### 3.2.1. Teacher Model Sensitivity

The response threshold (μ) for setting the pseudo-labels is determined based on the teacher model’s detection sensitivity (ts) for the training data (see Equation (2)). In our proposed solution, ts was set as 90%, as the value is close to the maximum BM detection sensitivity the teacher model can achieve. In this analysis, μ values were determined by setting the teacher model’s detection sensitivity to 80% and 85% for the training bin; hence, the impact of more-specific yet less-sensitive pseudo-labeling was evaluated. The other parameters (i.e., λ=1 and a single student model iteration) were kept unchanged during the experiment. Figure 7 and Table 2 present the findings of this analysis.

#### 3.2.2. Unlabeled Data Weights (*λ*)

The analysis aimed to represent the impact of reducing pseudo-labeled data weight during the student model’s training. This was achieved by evaluating the student models derived with λ=0.8 and λ=0.6 (λ=1.0 is the default value, see Equation (3)). The other parameters were kept at their default values during the experiment (i.e., ts = 90% and a single student model iteration). Please see Figure 8 and Table 3 for the results.

#### 3.2.3. Student Model Iterations

The experiment was targeted to assess the effect of further student training iterations, which are not part of our default solution due to low expected performance gains with a high additional computational cost. More explicitly, the student model from iteration-n (i.e., $\theta_{n}^{s}$) was used as the teacher model in the next iteration; $\theta_{n}^{s}$ becomes $\theta_{n+1}^{t}$ and generates pseudo-labeled data for the training of $\theta_{n+1}^{s}$. Two additional training and testing iterations were performed for the experiment (i.e., the student model iterations 2 and 3), where the other default parameters were kept unchanged (ts = 90% and λ=1.0). Please see Figure 9 and Table 4 for the corresponding results.

#### 3.2.4. Labeled-Data Utilization

Finally, the teacher and student models with a reduced number of labeled training data were evaluated. First, a random selection of 75% of patients were kept in the training bins (i.e., 60 patients), and the teacher and student models were trained using the labeled data collected from these patients and the complete set of unlabeled data. The test bins were kept unchanged, and the default noisy-student parameters were used during the analysis (i.e., λ=1, ts = 90%, and a single student model iteration). Next, the experiment was repeated by randomly keeping 50% of patients in the training bins (i.e., 40 patients). Figure 10 and Table 5 present the findings of this experiment.

## 4. Discussion

The validation study showed that the sensitivity-based NS training improves the framework performance by reducing the AFP count (i.e., 9.23 to 8.44 at 90% and 5.74 to 4.82 at 85% BM detection sensitivities). The algorithm’s parameters need to be set properly to exploit its potential: (1) using the peak teacher model sensitivity (ts) for the training data to determine μ, (2) adopting λ=1, and (3) performing a single student training iteration led to satisfactory results in our experiments; the motivations for each of these choices and relevant experiments are provided.

The first experiment showed that reduced teacher model sensitivity during pseudo-labeling (i.e., yielding more-accurate yet less-sensitive pseudo-labeled data) leads to higher AFP counts. As shown in Table 2, setting ts value to 80% caused the framework to produce the AFP value of 10.84 at 90% detection sensitivity (AFP values for 80% and 85% detection sensitivities for this setup were also relatively higher compared with the results acquired for ts = 90%). The NS-based training’s effectiveness is partly due to the implicit data augmentation/enrichment it introduces. Accordingly, one may argue that keeping the teacher model highly sensitive allowed student model training with pseudo-labeled data, representing high variability. Thus, this higher level of implicit data augmentation led to an improved BM detection performance.

The second experiment aimed to find the effect of reducing the weight of pseudo-labeled data during the student model training. Two alternative outcomes for this analysis were hypothesized before the experiment: (1) *λ* value behaves as an interpolator parameter between the teacher and student models, and (2) a reduced *λ* (<1.0) may cause the student model to outperform the default setup (using *λ* = 1.0) by limiting the drift towards pseudo-samples, which have limited accuracy. The analysis’ outcome supported the first of these (i.e., interpolation hypothesis). Table 3 shows that the students generated with *λ* = 0.6 and *λ* = 0.8 had performances bounded by the teacher and default student models, where the *λ* = 0.6 performed closer to the teacher model and *λ* = 0.8 performed closer to the default student model, respectively. Thus, the experiment’s result also complements the first experiment by supporting the notion that a higher amount of novel information contributes to a better student model performance.

The third experiment examined the student model’s performance with further student training iterations. The highest framework performances were observed after the first and second student model iterations, whereas the performance did not improve significantly after the third iteration (see Table 4). Our finding is consistent with [12], showing that the student model performance converges to a value at an arbitrary student model iteration, beyond which the accuracy does not change noticeably.

The final experiment aimed to quantify the impact of labeled training data amount on the framework’s accuracy. Accordingly, the teacher/student model training and validation procedures were repeated using reduced percentages of the labeled data (i.e., 75% and 50%). The results show that the amount of labeled data is critical to the final performance; the student models produced 8.44, 10.79, and 12.37 false BM detections by utilizing 100%, 75%, and 50% of the labeled training data. Furthermore, a valuable performance improvement with the use of proposed NS-based training was observed; it led to ~9% (from 9.23 to 8.44), ~12% (from 12.19 to 10.79), and ~11% (from 13.89 to 12.37) reduction in AFP counts in these experiments (see Table 5).

Table 6 presents comparative information between a group of SOTA BM detection frameworks (i.e., [3,4,5,6,7,8,33]) and the proposed one. It provides (1) study patient counts, (2) neural network architectures, (3) data acquisition types, (4) dimensional information on detected tumors, (5) data partitioning during validation studies, and (6) test performances with regards to AFP and sensitivity. Please note that the table includes results from [4] that used a similar dataset during its validation as this study. The data show that the potential advantages of our framework include (1) the detection of particularly small BM lesions with relatively low AFP and high sensitivity and (2) the capability to utilize unlabeled datasets during its model training via NS-based strategy.

## 5. Conclusions

In this study, a novel formulation of the NS-based self-training strategy, which is applicable for detection systems prioritizing low false-positive counts for arbitrary detection sensitivities, was introduced. Next, the BM detection framework for 3D T1c data was extended with the introduced strategy. The analyses showed that the method reduced AFP by 9% (from 9.23 to 8.44) for 90% tumor-detection sensitivity by utilizing 1247 unlabeled T1c exams. Future studies may (1) further validate the approach with larger and multi-institutional unlabeled datasets and (2) investigate the algorithm’s integration into other medical applications, such as the detection of primary brain tumors [35].

The noising procedure is essential in NS-based training strategies as it enables knowledge expansion. In data-limited clinical applications, its importance becomes even more prominent: the utilized highly parametric models could greatly benefit from the process, which provides data and model noise, to reduce the overfitting and hence improve their generalizability [36]. A wide range of noising mechanisms can be employed during the NS-based training, and the most appropriate selection may depend on (1) the data (i.e., type, amount, and characteristics) and (2) the deployed ML models. For instance, [12] utilized the stochastic depth approach for model noising [37], which is applicable for very deep neural networks (e.g., [38]), unlike the CropNet. The impact of a set of pertinent mechanisms such as (1) Mixup [16], (2) constrained GAN ensembles [39], and (3) stochastic dropouts [40] represent potential avenues for future research on the model noising aspect of the framework.

The applicability of artificial intelligence (AI) in medicine (among other fields) has increased significantly due to advancements in ML over the recent years. Accordingly, there have been various efforts to integrate AI in radiology workflows [41,42], where the AI-based medical imaging algorithms are deployed and updated periodically by utilizing a constant flow of annotated/labeled data. These workflows may also be adapted to use unlabeled data by adopting self-training methodologies, such as the one described in this study. Hence, the deployed medical imaging models are highly relevant to radiology AI systems that could benefit from the massive collection of unlabeled data sets stored in PACS systems.

## Figures and Tables

**Figure 1 diagnostics-12-02023-f001:**
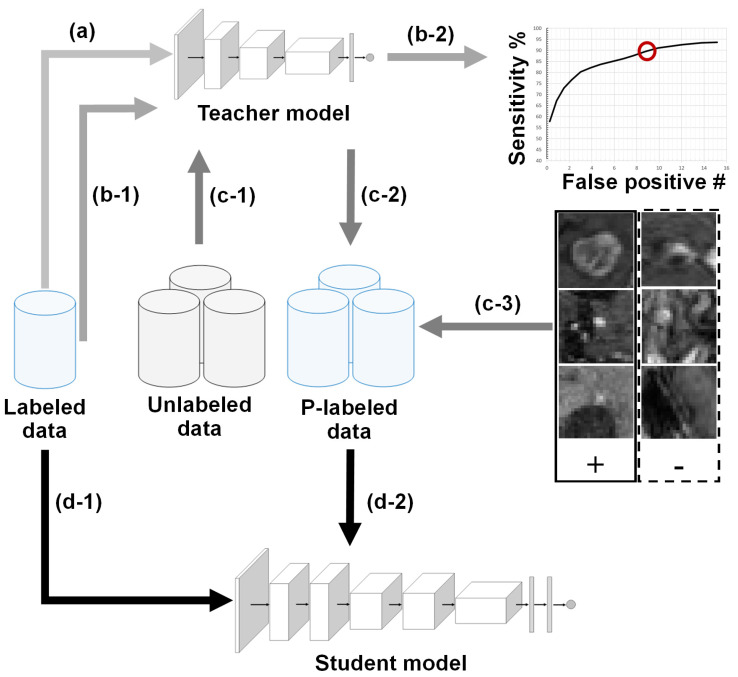
Adaptation of the NS-based self-training for the BM detection framework. The labeled T1c data is used for the training of a lower-capacity teacher model (a). Based on the labeled data (b-1) and a target sensitivity (b-2), the teacher model’s threshold for pseudo-labeling is determined. Via this threshold and unlabeled data (c-1), the teacher model generates paired pseudo-labeled data(c-2). The higher-capacity student model is trained using labeled (d-1) and pseudo-labeled (d-2) T1c data. The training of teacher and student models could be performed iteratively, where the final student model is utilized as the deployment model.

**Figure 2 diagnostics-12-02023-f002:**
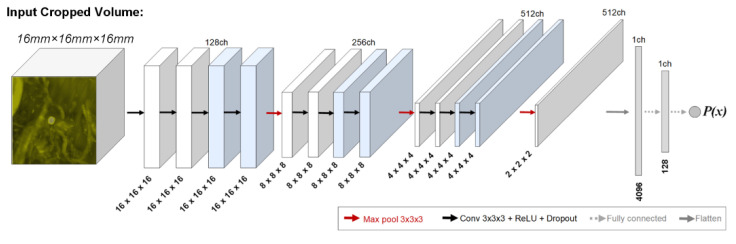
Framework-specific teacher–student models. The network architectures of CropNet-b4-16mm (all blocks) and CropNet-b2-16mm (blocks excluding the blue ones). The input is a 16 mm × 16 mm × 16 mm isotropic region of interest (ROI), where the output is a scalar in the range of [0, 1] giving the probability of input ROI including a BM lesion. The proposed solution utilizes CropNet-b2-16mm (lower-capacity) as the teacher and CropNet-b4-16mm (higher-capacity) as the student models.

**Figure 3 diagnostics-12-02023-f003:**
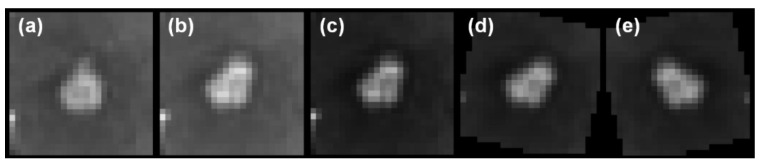
The data noising process. (**a**) Mid-axial slice of an original cropped sample. (**b**) Random elastic deformation, (**c**) random gamma correction, (**d**) random image rotation, and (**e**) random image flip are applied.

**Figure 4 diagnostics-12-02023-f004:**
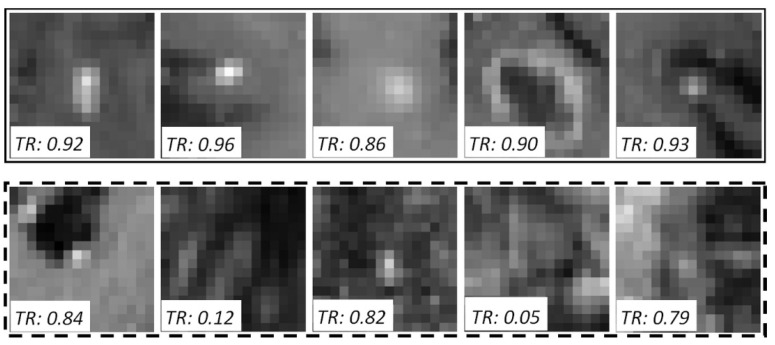
Mid-axial slices of an example pseudo-labeled batch of five pairs. The upper and lower rows show the positive and negative samples respectively. The response threshold (μ) is 0.85, and the teacher model responses (TR = $\theta^{t}\left({\hat{x}}_{i}\right)$) are given with the images.

**Figure 5 diagnostics-12-02023-f005:**
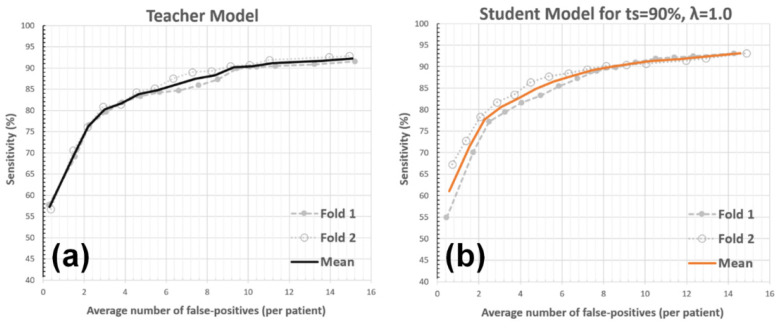
AFP versus detection sensitivity. The average number of false positives per patient (i.e., wrongly detected BM lesions for each patient) in relation to the sensitivity is illustrated for the teacher (**a**) and student (**b**) models. The mean curve (shown as the thick-line curve) represents the average of the CV folds.

**Figure 6 diagnostics-12-02023-f006:**
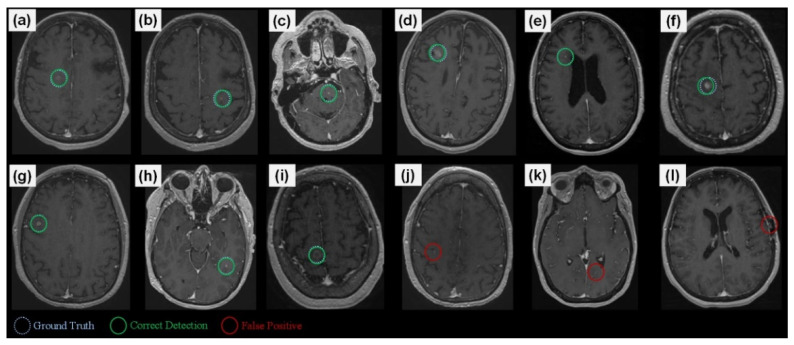
A set of detection examples. (**a**–**i**) Correct BM detections, (**j**,**k**) small vessels are wrongly detected as BMs, and (**l**) a formation in a surgical region is detected as a BM. The framework processes a given exam by first generating candidates and then processes these via CropNet, where the high response candidates are presented as the detection results.

**Figure 7 diagnostics-12-02023-f007:**
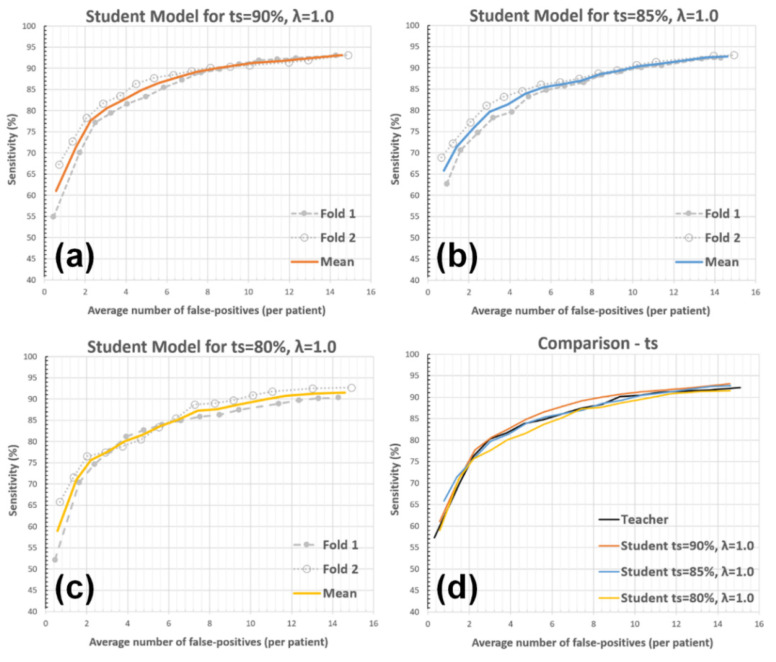
The impact of response threshold (μ) on the AFP metric. The student model performances for (**a**) ts = 90%—(default setup), (**b**) ts = 85%, (**c**) ts = 80%, and (**d**) comparisons between the teacher and student models with these setups are presented.

**Figure 8 diagnostics-12-02023-f008:**
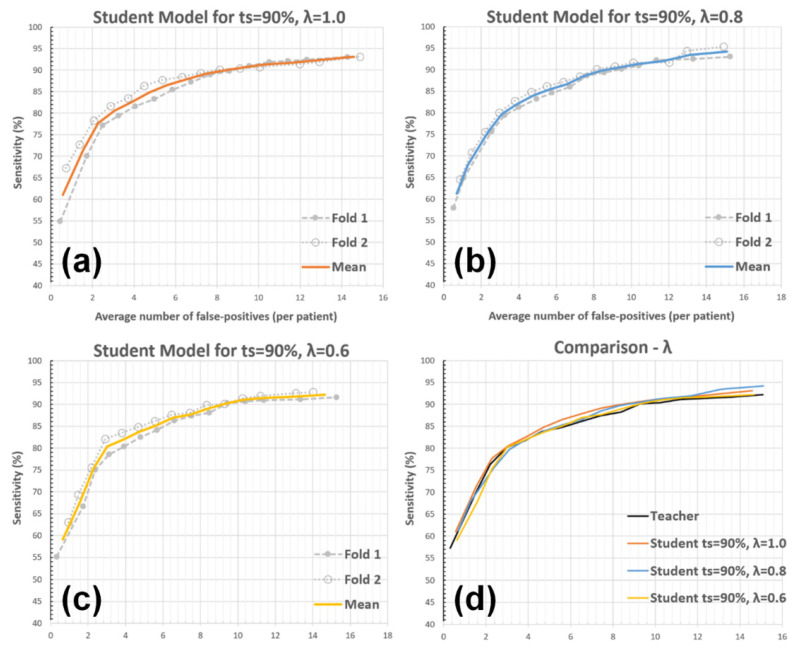
The impact of *λ* on the AFP metric. The student model performances for (**a**) *λ* = 1.0 (default setup), (**b**) *λ* = 0.8, (**c**) *λ* = 0.6, and (**d**) comparisons between the teacher and student models with these setups are presented.

**Figure 9 diagnostics-12-02023-f009:**
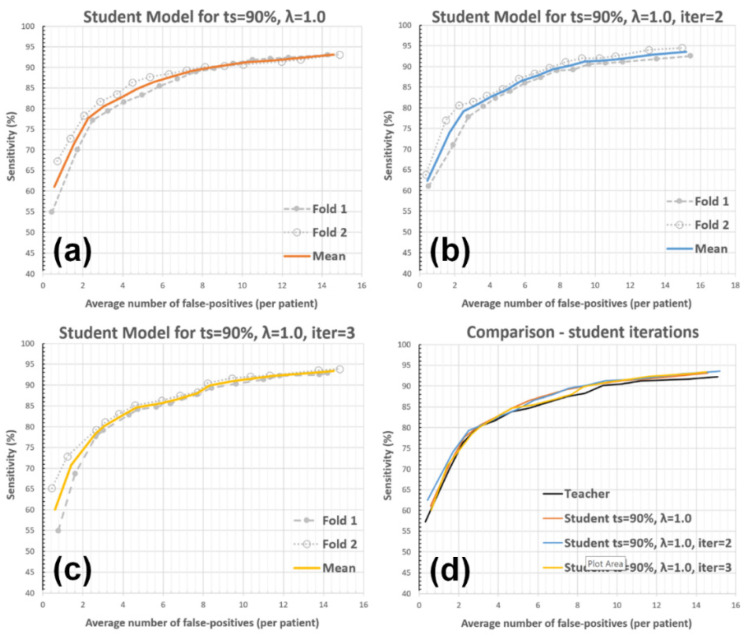
The impact of student model training iterations on the AFP metric. The student model performances for (**a**) single training iteration (default setup), (**b**) two iterations, (**c**) three iterations, and (**d**) comparisons between the teacher and student models with these setups are presented.

**Figure 10 diagnostics-12-02023-f010:**
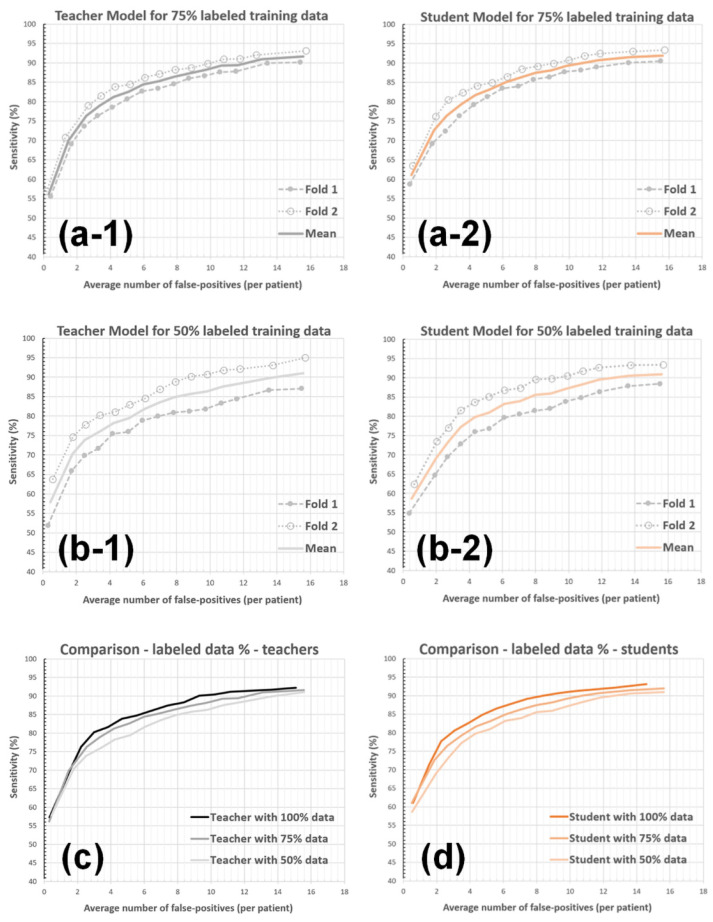
The impact of the amount of labeled data on the AFP metric. The teacher (**a-1**) and student (**a-2**) models were trained using 75% of the labeled data. The teacher (**b-1**) and student (**b-2**) models were trained using 50% of the labeled data. The teacher (**c**) and student (**d**) models were compared for different labeled-data utilization percentages.

**Table 1 diagnostics-12-02023-t001:** Labeled and unlabeled data summary.

	Labeled Data	Unlabeled Data
Exam count	217	1247
Patient count	158	867
Acquisition date from	January 2015	November 2016
Acquisition date to	February 2016	December 2019
Mean patient age	62	56
Sex ratio	0.89	1.09

**Table 2 diagnostics-12-02023-t002:** AFP vs. sensitivity for different ts values.

Detection Sensitivity	Teacher	Student ts = 90%	Student ts = 85%	Student ts = 80%
80%	2.94	2.91	3.18	3.82
85%	5.74	4.82	5.41	6.37
90%	9.23	8.44	9.97	10.84

**Table 3 diagnostics-12-02023-t003:** AFP vs. sensitivity for different λ values.

Detection Sensitivity	Teacher	Student *λ* = 1.0	Student *λ* = 0.8	Student *λ* = 0.6
80%	2.94	2.91	3.19	2.97
85%	5.74	4.82	5.36	5.55
90%	9.23	8.44	8.71	9.12

Results with ts = 90% and a single student model iteration. The gray column corresponds to the default setup.

**Table 4 diagnostics-12-02023-t004:** AFP vs. sensitivity for different student model iterations.

Detection Sensitivity	Teacher	Student Iter = 1	Student Iter = 2	Student Iter = 3
80%	2.94	2.91	2.89	3.03
85%	5.74	4.82	5.20	5.09
90%	9.23	8.44	8.35	8.46

Results with ts = 90% and λ=1.0. The gray column corresponds to the default setup.

**Table 5 diagnostics-12-02023-t005:** AFP vs. sensitivity for different amount of labeled-data (LD) utilization percentages.

Detection Sensitivity	Teacher LD:100%	Student LD:100%	Teacher LD:75%	Student LD:75%	Teacher LD:50%	Student LD:50%
80%	2.94	2.91	3.74	3.73	5.36	4.46
85%	5.74	4.82	6.60	6.15	8.05	7.65
90%	9.23	8.44	12.19	10.79	13.89	12.37

Results with ts = 90%, *λ* = 1.0, a single student model iteration. The gray column corresponds to the default setup.

**Table 6 diagnostics-12-02023-t006:** Overview of selected group of SOTA BM segmentation/detection approaches.

Study	Patient #	Network	Acq.	BM Volume (mm^3^)	Validation Type	Sens %	AFP
Charron et al. [3]	182-labeled	DeepMedic	Multi seq. ^a^	Mean: 2400 Median: 500	Fixed train/test	93	7.8
Liu et al. [5]	490-labeled	En-DeepMedic	Multi seq. ^b^	Mean: 672	5-fold CV	NA	NA
Bousabarah et al. [8]	509-labeled	U-Net	Multi seq. ^c^	Mean: 1920 Median: 470	Fixed train/test	77–82	<1
Grøvik et al. [6]	156-labeled	GoogleNet	Multi seq. ^d^	NA	Fixed train/test	83	8.3
Cao et al. [7]	195-labeled	Asym-UNet	T1cMRI	NA ^g^	Fixed train/test	81–100 ^k^	NA
Zhou et al. [33]	266-labeled	Custom + SSD	T1c MRI ^e^	NA ^h^	Fixed train/test	81	6
Dikici et al. [4]	158-labeled ^i^	CropNet	T1c MRI	Mean: 160 Median: 50	5-fold CV	90	9.12
This study	158-labeled 867-unlabeled	CropNet+NS	T1c MRI ^f^	Mean: 160 ^j^ Median: 50	2-fold CV	90	8.44

^a^ T1-weighted 3D MRI with gadolinium injection, T2-weighted 2D FLAIR, and T1-weighted 2D MRI sequences. ^b^ T1c and T2-weighted FLAIR sequences. ^c^ T1c, T2-weighted, and T2-weighted FLAIR sequences. ^d^ Pre- and post-gadolinium CUBE, post-gadolinium T1-weighted 3D axial IR-prepped FSPGR (BRAVO), and 3D CUBE FLAIR sequences. ^e^ 3D T1-weighted contrast-enhanced spoiled gradient-echo MRI sequence. ^f^ The same dataset was used for training and validation in this study and [13]. ^g^ Tumor volumes were not reported. The testing was conducted for 72 smaller tumors (1–10 mm in diameter) and 17 larger tumors (11–26 mm in diameter) separately. ^h^ Tumor volumes were not reported. The average size of tumors in the study were 10 mm ± (standard deviation: 8 mm). ^j^ The volumetric stats are for the labeled data. ^k^ Sensitivity was 81 percent for smaller tumors and 100 percent for larger ones. ^i^ The labeled dataset used in the study is similar with the one used in this current study.

## Data Availability

The data is not available for public access.

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
