# Peer review of "Advancing Brain Metastases Detection in T1-Weighted Contrast-Enhanced 3D MRI Using Noisy Student-Based Training"

_diagnostics, 2022, doi:10.3390/diagnostics12082023_

Round 1

Reviewer 1 Report

In this study,  a new formulation of the NS-based self-training strategy is introduced, which is applicable for detection systems prioritizing low false-positive counts for arbitrary detection sensitivities. Then, the authors extend their Brain Metastases detection framework for 3D  T1c data with the introduced strategy.

The article is well written, and the results were extensively discussed. However, comparing the work that was extended with the same data used by the previous one (taking advantage of the fact that they are the same authors) is recommended. 

The above is important in order to determine the real importance of what has been added to this new work. 

Author Response

Regards

Reviewer 2 Report

Dear authors,

The work is good and the results are appropriate.

I request you to consider the following suggestions:

Major:

a. The quality of images is very poor and it can be improved.

b. I request you to avoid using I/We in the manuscript and replace it with an appropriate word. 

General suggestions:

1. Please discuss the need for the data noising process. (For academic research, it is fine. For the real clinical study, is it necessary? Please discuss). 

2. In Figure 6, the whole axial-plane slice is presented. In Figures 3 and 4, cropped images are presented. Kindly discuss the need for labelled data in this system and if a whole image is given, will it work? 

3.  Please include this article in reference: The 2016 World Health Organization Classification of Tumors of the Central Nervous System: a summary

4. Figure 1 can be improved further and the role of the teacher and student model can be discussed in the image itself (optional comment).

5. Subsection 2.4 Database provides an overview of the data. I request you to present it in Table form for easy verification. 

Author Response

Regards

Reviewer 3 Report

The authors propose a new framework for Matastases Detection using Deep Learning based on MRI images. Moreover, the authors propose a novel NS-based semi-supervised training setup introducing a pseudo-labeling strategy. 

The article is well structured and written. The methodology is appropriate, and the results are contrasted with those in the literature.

They face a challenging problem in medicine that is the lack of annotated images, and they use a semi-supervised framework to ease this problem.

To improve and for completeness of the manuscript, I suggest giving technical details of the Crop-Net model.

Give a note indicating whether in the separation of the data set in training and test the subjects were separated in such a way that the same subject does not appear in both sets.

Author Response

Regards
